# Development and Seasonal Variations of the Larvae of Three Mesopelagic Fishes near Coral Reefs in the Red Sea

**Mohamed Ahmed Abu El-Regal** [1,*] **and James G. Ditty** [2]

[1] Marine Biology Department, Faculty of Marine Science, King Abdulaziz University, Jeddah 21589, Saudi Arabia
[2] American Southwest Ichthyological Researchers (ASIR), Albuquerque, NM 87123, USA
* Correspondence: mabuelregal@kau.edu.sa

**Abstract:** This work aims to describe the larval stages and the seasonal variation in the abundance of three mesopelagic species whose larvae are surprisingly abundant near coral reef areas in the Red Sea. The larvae were collected monthly using a plankton net (500 μ) from three coastal coral reef areas surrounding Sharm El-Sheikh on the Egyptian Red Sea coast between January and December 2015. The identification of larvae was based on the morphological and meristic characteristics according to the available literature. The larvae of this species were divided into preflexion, flexion, and postflexion stages and they were also categorized according to their size into relevant size classes. Mesopelagic fishes were represented in the collection by four species belonging to four families: *Vinciguerria mabahiss* (Family: Phosichthyidae), *Benthosema pterotum* (Family: Myctophidae), *Astronesthes martensii* (Family: Stomiidae; subfamily: Astronesthinae), and *Trichiurus* sp. (Family: Trichiuridae). In general, a total of 3678 larvae were collected, of which 1191, constituting about 32% of the total fish larvae, belonged to mesopelagic species. The most abundant species was *V. mabahiss*, with 677 larvae that constituted 18% and 57% of the total larvae and mesopelagic fish larvae, respectively. The second most abundant species was *B. pterotum*, which was represented by 485 larvae (13% of the total larvae and 40% of the mesopelagic fish larvae). *A. martensii* was represented by the lowest number of larvae (29 larvae, 2%). Most larvae of the three species were small and in the preflexion stage, whereas larger larvae are absent. They are highly abundant in the cooler months of the year between November and April. The high number of preflexion larvae may indicate that the three mesopelagic species spawn in the colder times of the year.

**Keywords:** mesopelagic development; fish; larvae seasonal distribution; Red Sea

**Key Contribution:** The larvae of three mesopelagic fish are described of which two species are described for the first time. Data about the spatial and temporal abundance of these larvae may be an indication of the spawning seasons and grounds of the species under investigation. The length frequency distribution of the three mesopelagic species indicate reproduction events in the colder period of the year.

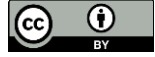

## 1. Introduction

Mesopelagic (or midwater) fish live in the intermediate pelagic water masses between the euphotic zone at 100 m and the deep bathypelagic zone at 1000 m where no light is found [1,2]. Most mesopelagic species undergo vertical migration into the epipelagic zone at night to feed and later return to mesopelagic depths during the day [3].

Mesopelagic species are the most abundant marine vertebrates in the world oceans [4] with a total biomass of billions of tons, of which about 300 million tons occur in the Indian Ocean. These estimates represent approximately 10 times the biomass of the world's total fish catch. Due to the high proportion of wax esters of limited nutritional value, however, only a few species, such as *Benthosema glaciale* in the Oman Sea [5], are

potentially abundant enough for commercial purposes. Although most species lack commercial importance, mesopelagic fishes play essential roles in the ocean's pelagic and benthic food webs and are able to affect the dynamics of their zooplankton prey [6,7] and predators, such as some commercially important pelagic fishes [8], marine mammals [9], seabirds [10], and benthic predators [11], and are important in transferring energy between ocean ecosystems [11].

Due to their small size, mesopelagic fish are characterized by low fecundity that range from a few hundred to a few thousand eggs over the spawning period. Most species are oviparous with plankton eggs and larvae, and some exhibit batch-spawning characteristics (i.e., spawn eggs in batches over several months). Eggs are released either during the day in mesopelagic waters or at night in the epipelagic zone, where larvae hatch and may remain until metamorphosis, during which the light organs or photophores develop; thereafter, they return to deeper water juvenile and adult habitats. Some mesopelagic species are distributed worldwide, and many are circumpolar, especially in the Southern Hemisphere [12].

Much of the research on the distribution and natural history of mesopelagic fishes was conducted in the 1970s, when FAO (Food and Agriculture Organization) searched for new unexplored commercial resources. Interest in mesopelagic fishes has been renewed since the 1990s after the discovery that the sound-scattering layers (SSLs) in the ocean has high densities of mesopelagic species, which formed the basis for studies of the life history and adaptations of mesopelagic fish in the context of general ecological theory.

Investigations of the midwaters of the Red Sea have shown that the deep-water fauna in general is quite limited due to the unusually isothermal structure of the waters below the sea level, which makes the Red Sea unsuitable for all but a very few adaptable species [13].

Sixteen species of mesopelagic fish occur in the Red Sea (Table 1), with little being known about their biology and ecology. This work aims to describe the larval stages of three mesopelagic species that are abundant near coral reefs in the Red Sea. Larvae of two species, *V. mabahiss* and *A. martensii*, are described for the first time.

**Table 1.** Orders, families, and species of mesopelagic fish in the Red Sea (Golani and Fricke [14]).

| Order | Family | Species |
|---|---|---|
| Stomiiformes | Phosichthyidae | *Vinciguerria mabahiss* (Johnson and Feltes 1984) |
| | Stomiidae | |
| | Subfam: Astroneshthinae | *Astronesthes martensii* (Klunzinger 1871) |
| | Subfam: Stomiinae | *Chauliodus sloani* (Bloch and Schneider 1801) |
| | | *Stomias affinis* (Günther 1887) |
| | Sternoptychidae | *Maurolicus mucronatus* (Klunzinger 1871) |
| Myctophiformes | Myctophidae | *Benthosema fibulatum* (Gilbert and Cramer 1897) |
| | | *Benthosema pterotum* (Alcock 1890) |
| | | *Diaphus coeruleus* (Klunzinger 1871) |
| Ateleopodiformes | Ateleopodididae | *Ateleopus japonicus* (Bleeker 1853) |
| Scombriformes | Trichiuridae | *Evoxymetopon moricheni* (Fricke, Golani, and Appelbaum-Golani 2014) |
| | | *Tentoriceps cristatus* (Klunzinger 1884) |
| | | *Trichiurus auriga* (Klunzinger 1884) |
| | | *Trichiurus lepturus* (Linnaeus 1758) |
| | Gempylidae | *Thyrsitoides marleyi* (Fowler 1929) |
| Aulopiformes | Paralepididae | *Lestidiops jayakari* (Boulenger 1889) |
| | | *Lestrolepis luetkeni* (Ege 1933) |

## 2. Materials and Methods

### 2.1. Study Area

The present study was conducted at three coastal sites along the Gulf of Aqaba coast (Figure 1), Naama Bay (NAM), Sharm El-Maya Bay (SMB), and Port Bay (PRT), each with different ecological conditions [15]. Naama Bay is located approximately 15 km south of the Strait of Tiran at 27°55′ N and 34°20′ E, with a maximum depth of 100 m, and is exposed to touristic stress from its many resorts, sailing, and diving activities. This bay is bordered by fringing reefs characteristic of the Red Sea. Sharm El-Maya Bay is a semi-enclosed bay at 27°51.8′ N and 34°18.1′ E and is divided into two water bodies, a small, near-shore, sub-tidal area (mean depth: 9 m) and a larger area with a maximum depth of 90 m connected to the Red Sea through an opening that is 200 m wide. The bottom is covered mainly with seagrass patches. Port Bay lies at the entrance of Sharm El-Sheikh City at 27°51′ N and 34°16.7′ E, with a maximum depth of 35 m, and is subjected to less stress by tourism and other human activities than the other two sites. The borders of this bay are characterized by coral reefs, whereas the bottom is sandy with many reef and seagrass patches (Table 2).

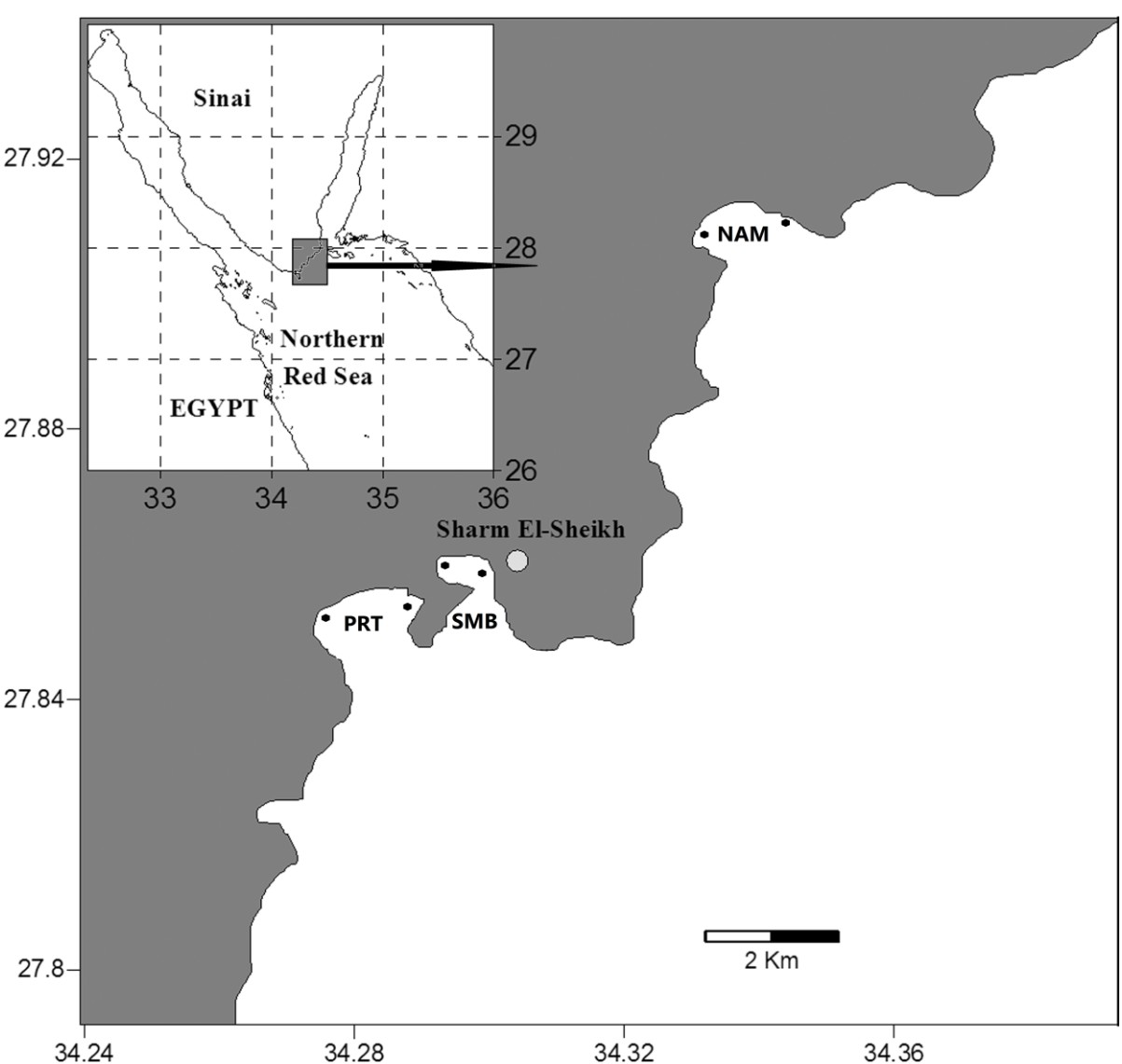

**Figure 1.** Map of the northern Red Sea and the sampling sites: Naama Bay (NAM), Port Bay (PRT), and Sharm El-Maya Bay (SMB) (after El-Sherbiny et al. [15]).

**Table 2.** Sites, codes, and habitat description of the surveyed sites: Naama Bay (NAM), Port Bay (PRT), and Sharm El-Maya Bay (SMB) in the Sharm El-Sheikh area.

| Site Name | Code | Coordinates | Habitat | Depth |
|---|---|---|---|---|
| Naama Bay | NAM | 27°55′ N and 34°20′ E, | Mainly coral reef | 100 m |
| Port Bay | PRT | 27°51′ N and 34°16.7′ E | Mainly coral reef | 35 m |
| Sharm El-Maya | SMB | 27°51.8′ N and 34°18.1′ E | Mainly seagrass | 10 m |

### 2.2. Field Work

Temperature, salinity, and pH were measured simultaneously using a portable multiparameter instrument (Aqua-Read Ap5000, Aquaread, Kent, England). Each site was sampled with a 100 cm mouth diameter and 500 μm mesh plankton net. The net was hauled horizontally parallel to the reef, with the top of the net at the water's surface for 10 min at a fixed speed of 2 knots [16,17]. Two stations in each bay were sampled monthly with three replicates from January to December 2015, which resulted in the collection of 216 plankton samples. The sampling process was conducted in the morning just before sunrise. A flowmeter fitted in the mouth of the net was used to estimate the volume of water filtered. Samples were fixed in buffered 5% formalin solution in seawater on board the ship and later preserved in 90% ethanol for further examination in the laboratory.

### 2.3. Laboratory Work

The samples were examined under a stereomicroscope, and larvae were sorted and separated by family. The larvae of mesopelagic fishes were separated, counted, and measured under the stereomicroscope with an eyepiece micrometer. Measurements were expressed as proportions of the body length (BL) and rounded to the nearest 0.1 mm. In this case, we considered BL to be equivalent to the standard length (SL) and used both interchangeably. Larvae were identified to the species level, where possible, using the relevant identification guides [18], and were divided into preflexion, flexion, and postflexion states. The larvae were placed in separate, labeled vials by species and life stage and preserved in 90% ethanol for further examination and description. The different stages of the mesopelagic larvae were observed under the dissection stereomicroscope (M 80 Leica, Leica Microsystems' instruments, Wetzlar and Mannheim, Germany) using camera Lucida.

### 2.4. Data Analysis

Univariate statistics were conducted in SPSS 22 using analysis of variance (ANOVA) to determine differences in the number of individuals and number of species between months and sites. A two-way ANOVA was used to test if the interaction between month and site was statistically significant. All data were tested for homogeneity of variance and normal distribution. Where the samples were not homogeneous, data were either transformed or the non-parametric Kruskal–Wallis test was used [19,20]. Pearson's coefficient was calculated to test the correlation between the abundance of the three species. In addition, the correlation between the abundance of each mesopelagic species was tested against the environmental parameters in each bay over the year. Graphs were created in GraphPad Prism 8 and SPSS 22.

### 2.5. Ethical Statement

Samples were collected by plankton net, and they were only plankton samples for which there are no rules to follow.

## 3. Results

### 3.1. Environmental Conditions in the Study Area

The average surface water temperature in the three investigated bays varied from the winter minimum (22.7 °C) in February to the summer maximum (30.6 °C) in August. Among the sampling sites, the monthly difference was mostly between 0.1 and 0.7 °C, increased to 1.3–1.8 °C in March, May, July, and September. The surface salinity showed slight variations over the whole area of study, varying monthly between 40.2 psu in March and 40.9 psu in August. The pH values varied from 7.72 in April to 8.27 in February, with an average of 7.997 ± 0.196 for the whole area.

### 3.2. General Abundance of the Larvae of Mesopelagic Fish

The mesopelagic fish in the current study were represented by four species belonging to four families and three orders: Stomiiformes, *Myctophiformes*, and Scombriformes. The order Stomiiformes was represented by two species in two families (Phosichthyidae, *V. mabahiss*, and Stomiidae, *A. martensii*). The order Myctophiformes was represented by *B. pterotum* (Myctophidae). Scombriformes were represented by one family, Trichiuridae, and one species, possibly *T. auriga*. The larvae of mesopelagic fishes constituted about 32% of all larvae (1191 of 3678 total larvae collected). The most abundant mesopelagic species was *Vinciguerria mabahiss* with 677 larvae, which represents 18% of all fish larvae collected and 57% of larvae of mesopelagic species, followed by *Benthosema pterotum* with 485 larvae (13% of all larvae and 41% of all mesopelagic larvae) (Figure 2, Table 3). The larvae of *Astronesthes martensii* (n = 29) accounted for 2% of all mesopelagic larvae. A 5 mm trichiurid larva, possibly *Trichiurus auriga*, was also collected in Sharm El-Maya Bay during July, but was excluded from further analyses.

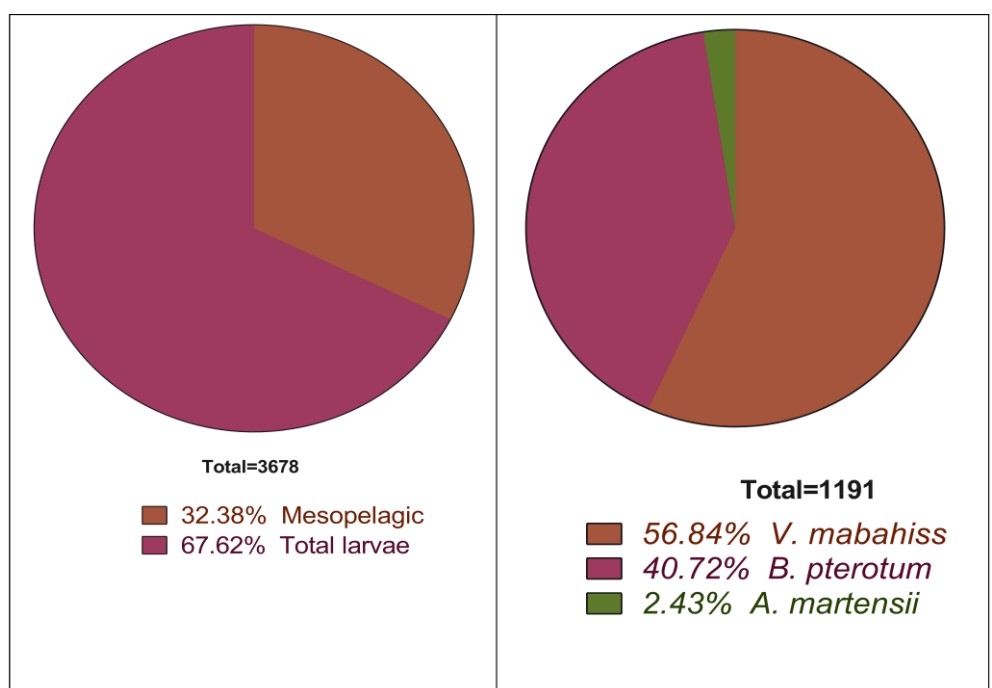

**Figure 2.** Percentage of the contribution of the mesopelagic larvae to the total fish larvae collected and of each mesopelagic species.

**Table 3.** Mean abundance of the larvae of mesopelagic fish from Sharm El-Sheikh during the sampling period from January to December 2015 (NAM, Naama Bay; PRT, Port Bay; SMB, Sharm El-Maya Bay).

| Month | *V. mabahiss* | | | *B. pterotum* | | | *A. martensii* | | |
|---|---|---|---|---|---|---|---|---|---|
| | NAM | PRT | SMB | NAM | PRT | SMB | NAM | PRT | SMB |
| Aug | 0 | 0 | 0 | 0 | 2 | 6 | 0 | 0 | 0 |
| Sep | 2 | 0 | 0 | 0 | 0 | 0 | 0 | 0 | 0 |
| Oct | 0 | 0 | 0 | 0 | 0 | 0 | 0 | 0 | 0 |
| Nov | 39 | 68 | 93 | 0 | 15 | 11 | 0 | 0 | 0 |
| Dec | 42 | 33 | 32 | 9 | 10 | 2 | 5 | 1 | 2 |
| Jan | 3 | 117 | 108 | 4 | 195 | 82 | 1 | 7 | 11 |
| Feb | 9 | 7 | 53 | 4 | 9 | 50 | 0 | 0 | 0 |
| Mar | 0 | 0 | 0 | 0 | 0 | 5 | 2 | 0 | 0 |
| Apr | 14 | 28 | 29 | 0 | 55 | 24 | 0 | 0 | 0 |
| May | 0 | 0 | 0 | 0 | 0 | 0 | 0 | 0 | 0 |
| Jun | 0 | 0 | 0 | 0 | 1 | 0 | 0 | 0 | 0 |
| Jul | 0 | 0 | 0 | 0 | 1 | 0 | 0 | 0 | 0 |
| **Sum** | **109** | **253** | **315** | **17** | **288** | **180** | **8** | **8** | **13** |

The ANOVA showed that the abundance of *V. mabahiss* differed significantly among months (F = 4.94, $p < 0.05$), but not among bays (F = 0.89, $p > 0.05$). The abundance of *B. pterotum* also differed significantly among months (F = 4, $p < 0.05$), but not among bays (F = 1.2, $p > 0.05$), as did *A. martensii* among months (F = 2.4, $p < 0.05$), but not among bays (F = 0.1, $p > 0.05$). Moreover, the interaction between site*month was not statistically significant.

There was a significantly positive correlation between the abundance of *V. mabahiss* and *B. pterotum* (r = 0.75; $p < 0.05$). Weak and moderate negative relationships were found between *A. martensii*, on the one hand, and *V. mabahiss* (r = −0.06) and *B. pterotum* (r = −0.4), on the other hand, respectively.

There was a moderately negative relationship between the abundance of the three species and the temperature, where the values of the Pearson's correlation coefficient were −0.44, −0.16, and −0.41 for *V. mabahiss*, *B. pterotum*, and *A. martensii*, respectively.

### 3.3. Vinciguerria Mabahiss

3.3.1. Description of the Larvae

The larvae are elongated with 37–39 myomeres and have a long gut (70–80% BL). The head has an elongated snout, large mouth, and elliptical eye; larvae also lack fin and head spines. The preflexion larvae of 5 mm have pectoral and pelvic fin buds, but lack anlage in the dorsal, anal, and caudal fins. The head length is about 20% BL, and the body depth was 8% BL at this size (Figure 3, Figure 4, Table 4). The notochord flexion occurs between 6 mm and 7 mm, after which the early postflexion larvae have a well-formed caudal fin. By 8 mm, rays begin to develop in the dorsal fin. By about 9 mm, rays begin to develop in the anal fin, which originates below the mid-base of the dorsal fin; canine teeth are also visible in both jaws at this size, but pectoral fin rays remain undeveloped. The larvae also have pigment at the base of the caudal fin (hypurals area), and three melanophores along the ventral margin of the caudal peduncle. By 12.5 mm, the dorsal and anal fins have 14 rays each, and rays develop in the pectoral fins. At this size, *V. mabahiss* typically have a melanophore above the cleithrum, one above the hindgut near the vent, one to three melanophores along the ventral margin of the caudal peduncle, and pigment dorsally above the caudal fin base.

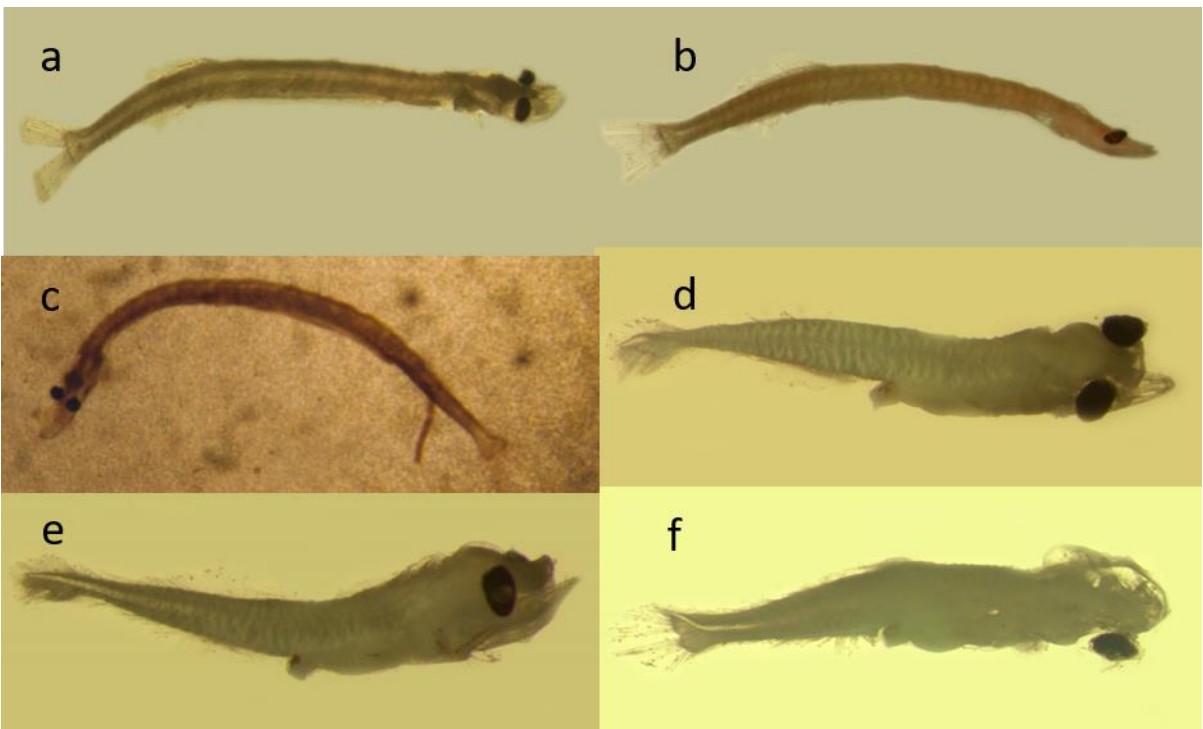

**Figure 3.** Photos of the three investigated mesopelagic species collected from the Sharm El-Sheikh area. (**a**,**b**) *V. mabahiss*; (**c**) *A. martensii*; and (**d**–**f**) *B. pterotum*.

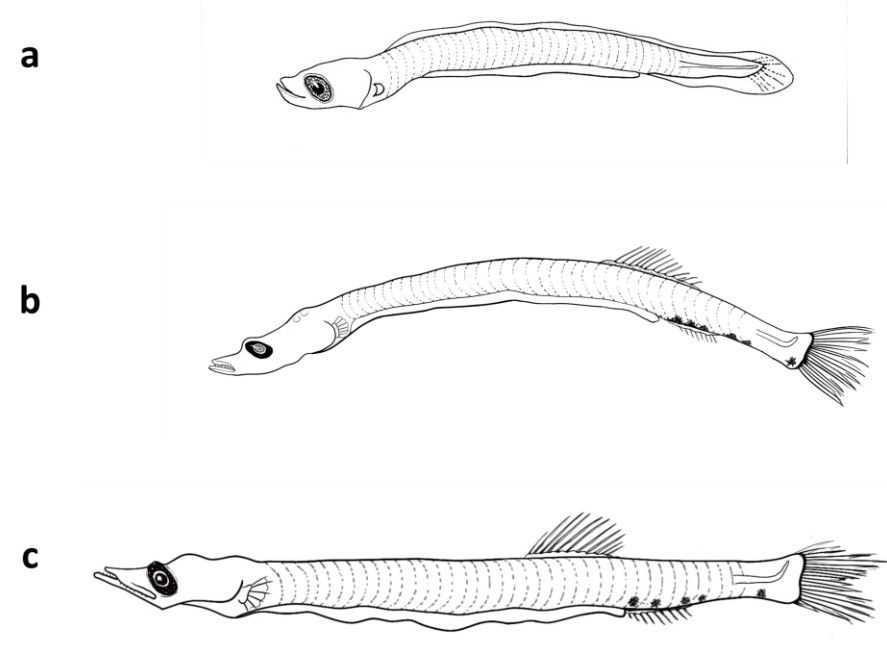

**Figure 4.** The developmental stages of the larvae of *V. mabahiss* collected from the Red Sea. (**a**) 5 mm, (**b**) 9 mm, and (**c**) 13 mm.

**Table 4.** Meristics of the mesopelagic species collected from the Red Sea.

| Meristics | *V. mabahiss* | *B. pterotum* | *A. martensii* |
|---|---|---|---|
| Myomeres | 37 | 30 | 50 |
| Preanal | 25 | 19 | 35 |
| Postanal | 12 | 11 | 15 |
| Doral fin spines | 0 | 0 | 0 |

| | | | |
|---|---|---|---|
| Doral fin rays | 13 | 12 | 10–21 |
| Anal fin spines | 0 | 0 | 0 |
| Anal fin rays | 13 | 18 | 12–22 |
| Pelvic | 7 | 8 | 5–9 |
| Pectoral | 10 | 19 | 5–9 |
| Caudal | 19 | 19 | NA |

The distinguishing characters of *V. mabahiss* include an elongated body; elliptical eyes and duck-like snout; the anal fin originates below the middle of the dorsal fin base; a myomere count of 37–39; and an unusual pigmentation pattern. Select morphometrics offer insight into changes in body morphometry by early life stage (Table 5).

**Table 5.** Morphometrics of the mesopelagic species collected from the Red Sea.

| | *V. mabahiss* | | *B. pterotum* | | *A. martensii* | |
|---|---|---|---|---|---|---|
| | **Preflexion** | **Postflexion** | **Preflexion** | **Postflexion** | **Preflexion** | **Postflexion** |
| Snl/HL | 37% | 33–40% | 33–40% | 25–30% | 25% | 30% |
| ED/HL | 25% | 22–26% | 33–40% | 16–20% | 18% | 35% |
| HL/BL | 26% | 16–20% | 20–25% | 25–26% | 16% | 36% |
| PAL/BL | 83% | 74–77% | 50–60% | 55–60% | trailing gut | trailing gut |
| PDL/BL | --- | 61–63% | --- | 44–50% | ---- | 61% |
| BD/BL | | | 13–15% | 15–17% | % | 6% |

Snl, snout length; ED, eye diameter; HL, head length; PAL, preanal length; PDL, predorsal length; BD, body depth; BL, body length.

The study of morphometrics showed that the ratios of head length (HL), body depth (BD, preanal length (PAL), and predorsal length (PDL) increase linearly with the standard length.

3.3.2. Abundance and Spawning Season of *V. mabahiss*

The larvae occurred in 39% of all samples collected and were the most abundant larvae of any mesopelagic species collected, with an overall abundance of 19/1000 m$^3$. The larvae occurred mostly during the cooler months from November to April, with peaks of abundance during November (200/1000 m$^3$) and January (228/1000 m$^3$). The larvae were rare in September (2/1000 m$^3$) and totally absent during the summer months (June–August) and in October and March (Figure 5). The larvae were collected at all sites sampled but were the most abundant in Sharm El-Maya Bay (315/1000 m$^3$) and the least abundant in Naama Bay (109/1000 m$^3$). The larvae reached their maximum abundance in Port Bay (PRT) in January (192/1000 m$^3$) and Sharm El-Maya Bay (SMB) in November (140/1000 m$^3$) (Figure 5).

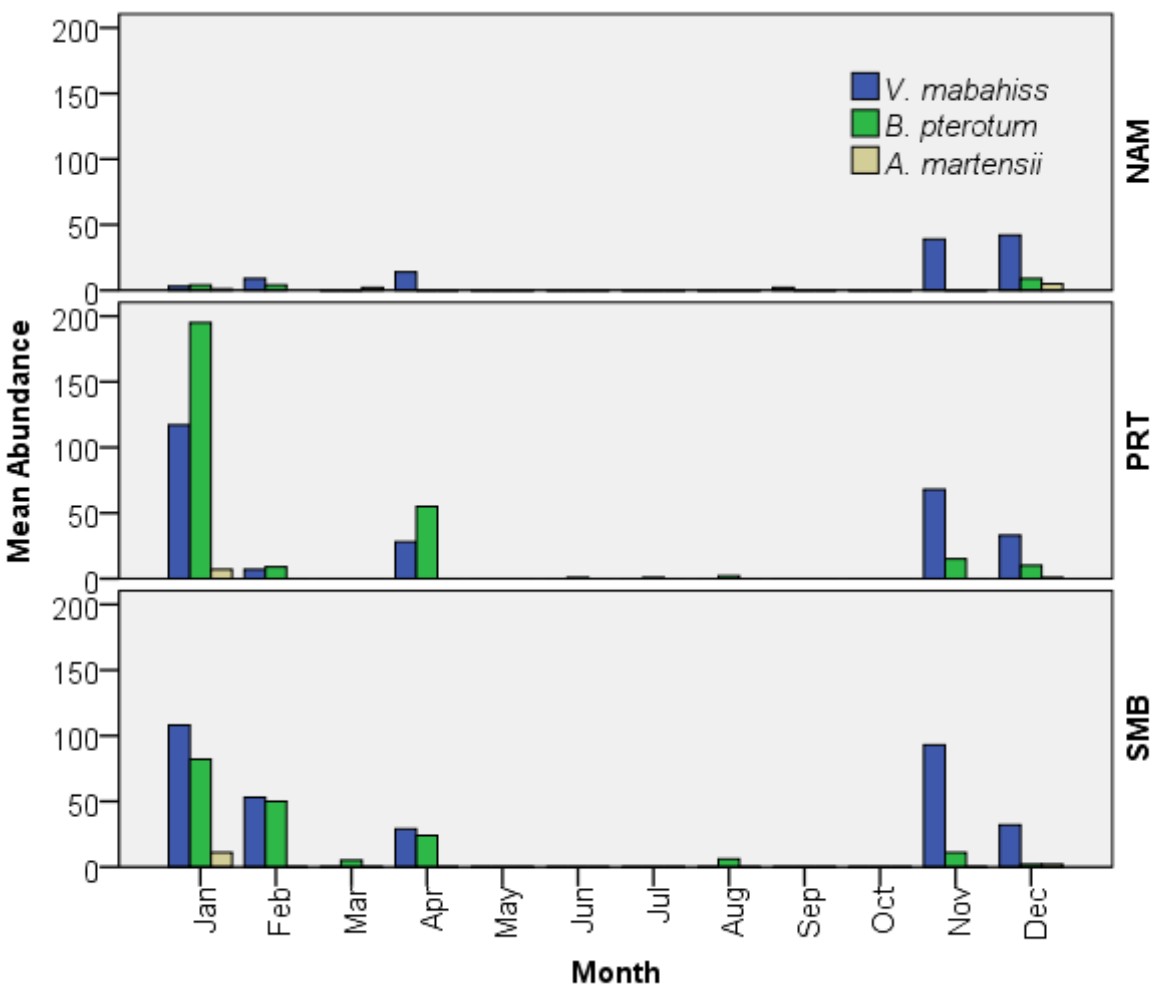

**Figure 5.** Mean abundance of the larvae of the three mesopelagic species collected at different bays: Naama Bay (NAM), Port Bay (PRT), and Sharm El-Maya Bay (SMB) from the Sharm El-Sheikh area in the period from January to December 2015.

### 3.3.3. Length–Frequency Distribution of *V. mabahiss* Larvae

The larvae of *V. mabahiss* occurred primarily from September to May with a total of 339 larvae collected (Figure 6). The *V. mabahiss* larvae ranged in size from 3 to 13 mm SL and were divided into six size classes with two-millimeter (mm) length intervals: I: 3–4.9 mm; II: 5–6.9 mm; III: 7–8.9 mm, IV: 9–10.9 mm, V: 11–12.9 mm; and VI: 13–14.9 mm. Overall, most larvae were at the preflexion stage (Figure 6) with size classes I–III (<8.9 mm) accounting for about 97% of all *V. mabahiss* larvae collected, while larvae > 8.9 mm only contributed 3%. Although the *V. mabahiss* larvae collected in November (n = 119) ranged in size from 3 to 10.9 mm SL, most were in size classes I or II, with one larva > 9 mm SL. In December, there were 66 larvae between 3 and 8.9 mm SL, but 48% were in the two smallest size classes, while January collections contained 86 larvae between 3 and 12.9 mm SL, with 48% in size class II (5–6.9 mm); 9% of *V. mabahiss* larvae collected in January were >8.9 mm SL (Figure 6, Figure 3). During February, 89% of all larvae were in size classes I and II (<6.9 mm), with two larvae being the largest between 13 and 14.9 mm SL. Of the 22 larvae collected in April between 3 and 10.9 mm SL, 86% were <4.9 mm SL. All larvae collected in May (n = 5) were <4.9 mm SL. The collection of larvae < 4.9 mm SL during all months in which *V. mabahiss* occur suggests that spawning may have occurred near the reef at those locations, mostly from November to April.

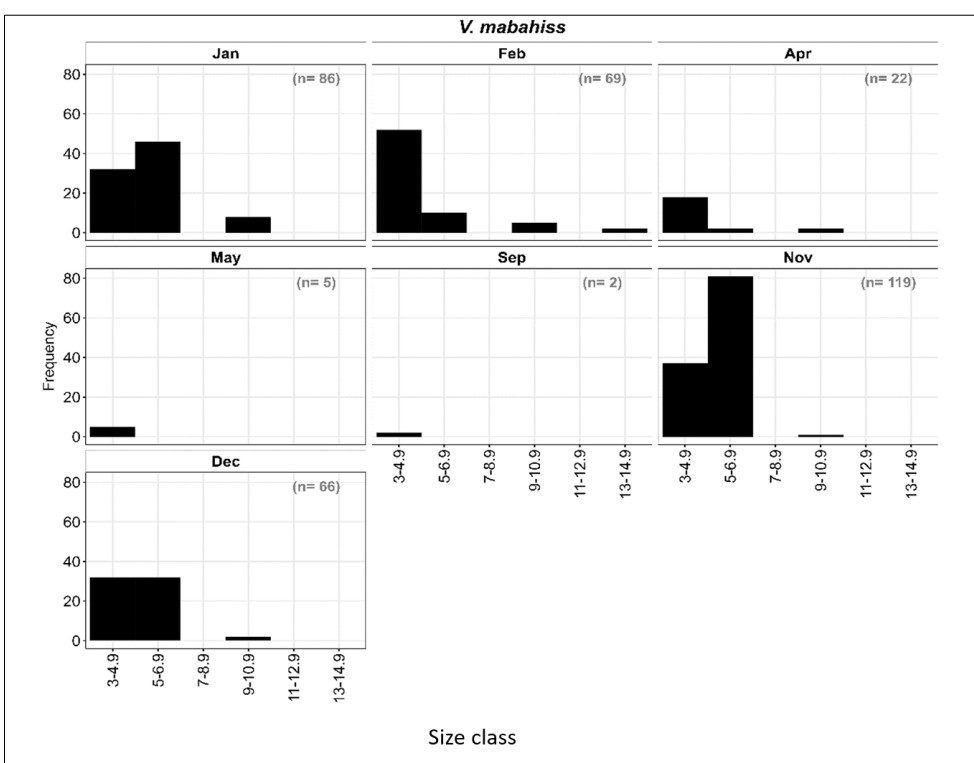

**Figure 6.** Length–frequency distribution of the *V. mabahiss* larvae at different months from the Sharm El-Sheikh area in the period from January to December 2015.

### 3.4. Astronesthes Martensii

3.4.1. Description of the Larvae

The larvae of *Astronesthes martensii* ranged in size from 6 to 13.5 mm SL, with all larvae < 8 mm at the preflexion stage. In general, the larvae of *A. martensii* are elongated and moderately slender with >50 myomeres, and have a small head with small elliptical eyes; both jaws have fang-like teeth. The long gut extends to about 70% SL and the posterior portion of the hindgut trails the body. The finfold along the dorsal and ventral margins of the body are peppered with small melanophores, and stellate, brown melanophores line the dorsal and ventral margins of the tail and extend forward onto the trunk. The head length is about 16% SL and the body depth about 6% SL. Notochord flexion begins at about 8 mm and is completed by 13.5 mm, when the caudal fin is well-formed and the dorsal and anal fins begin to form; however, the pectoral fins still lack rays. The anal fin originates well behind the dorsal fin and is slightly longer than the dorsal fin base (Figure 7, Figure 3). The larvae of *A. martensii* are distinguished from those of other mesopelagic fish in the Red Sea by the trailing gut, finfold pigmentation, relative position of the dorsal and anal fins, and a high myomere count.

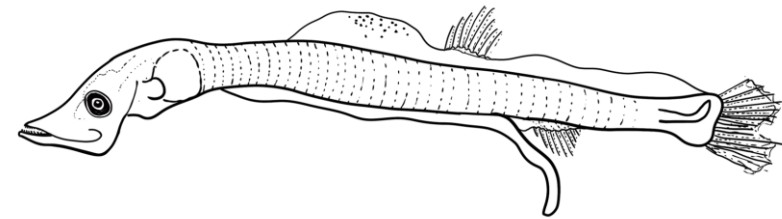

**Figure 7.** The larvae of *A. martensii* collected from Sharm El-Sheikh during the period from January to December 2015.

### 3.4.2. Abundance and Spawning Season of *A. martensii*

To our knowledge, information on the spawning seasonality of *A. martensii* has not been reported to date. All 29 larvae were collected during December, January, and March with a peak in January (Figure 5), whereas the March samples contained only two larvae. *A. martensii* accounted for <2% of all larvae of mesopelagic fishes, at a mean overall abundance of 1/1000 m³. Larvae occurred in all bays sampled but were most abundant in SMB (Figure 5). Even though the collections contained few larvae, our data suggest that *A. martensii* spawn during the winter months, although we cannot exclude the possibility that they may also spawn during other months and seasons.

### 3.4.3. Length–Frequency Distribution of *A. martensii*

The larvae were divided into four size classes at two mm length intervals: I: 6–7.9 mm; II: 8–9.9 mm; III: 10–11.9 mm; and IV: 12–13.9 mm SL. Most preflexion larvae were in the smallest size class. December collections contained 8 preflexion larvae, and 15 of 19 larvae were collected in January. Both larvae collected during March, however, were between 12 and 13.9 mm SL, which is also consistent with primary spawning during the winter months (December–January) (Figure 8).

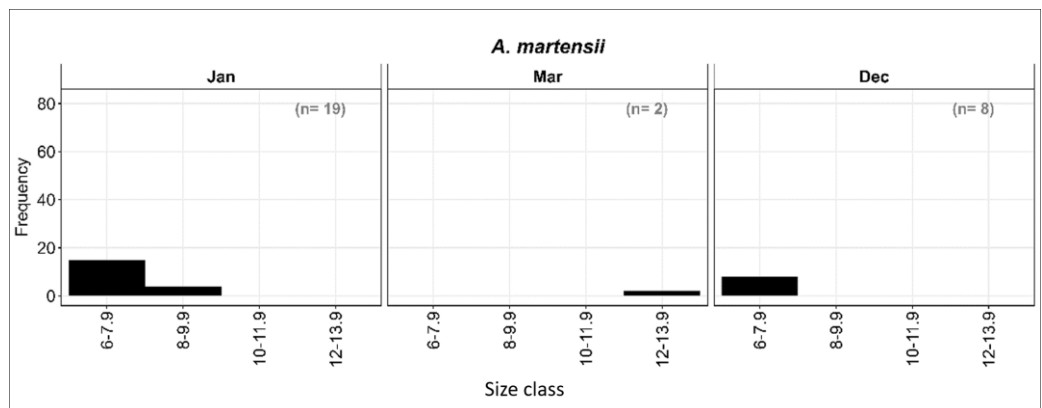

**Figure 8.** Length–frequency distribution of the *A. martensii* larvae at different months from the Sharm El-Sheikh area in the period from January to December 2015.

## 3.5. *Benthosema pterotum*

### 3.5.1. Description of the Larvae

The larvae of *B. pterotum* (Figure 9, Figure 3) have robust bodies, elliptical eyes with a small, lunate choroid mass ventrally, and a sigmoid-shaped gut with transverse mucosal folds that extend to mid-body. A gap between the anus and anal fin origin closes between 9 and 10 mm SL as the anal fin develops. The body becomes shorter and deeper as larvae develop, and the gut narrows posteriorly. The sequence of fin formation is as follows: pectoral—primary caudal—anal and dorsal—secondary caudal—pelvic fin rays. Larvae begin to transform at about 10 mm SL.

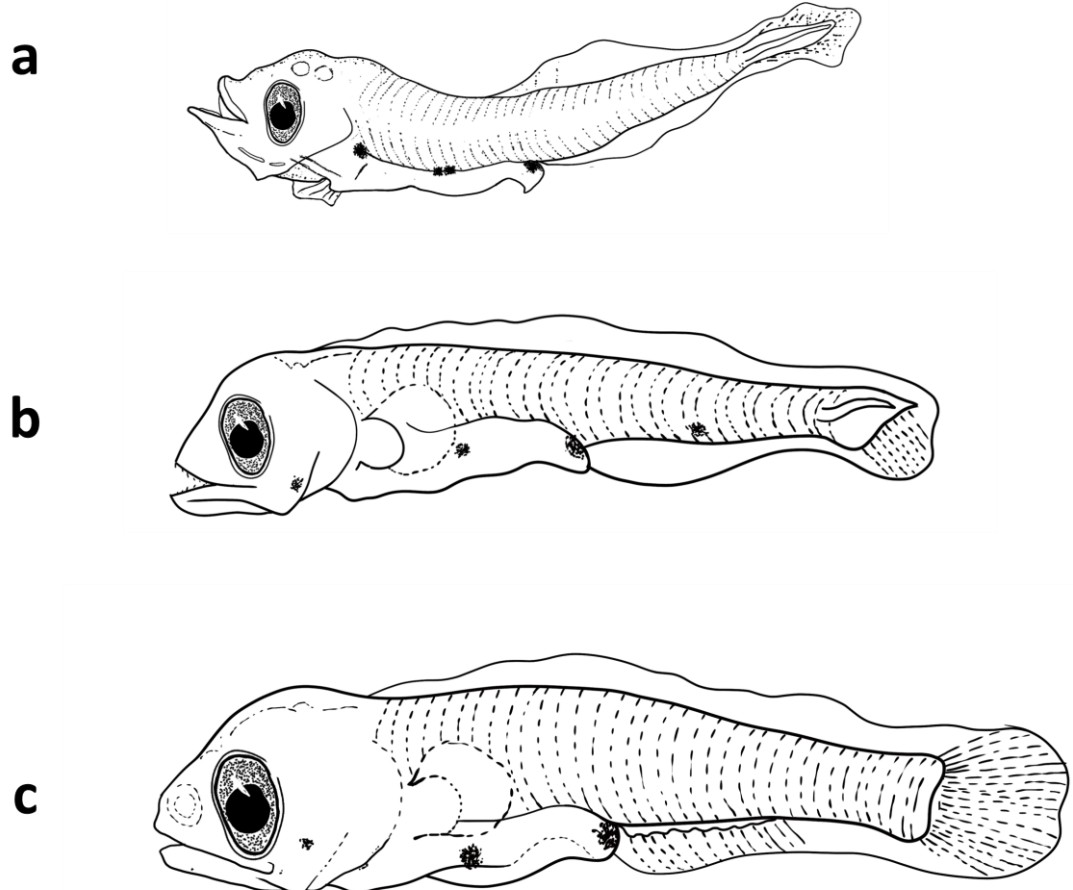

**Figure 9.** The larvae of *B. pterotum* larvae from the Sharm El-Sheikh area in the period from January to December 2015. (**a**) 2.1 mm, (**b**) 4 mm, and (**c**) 6 mm.

Small preflexion larvae have 12–15 preanal myomeres and 30–32 total myomeres. The gut terminates near the mid-body (preanal length about 55% SL), and the larvae have pigment along the ventral margin of the gut and over the dorsal margin of the vent. The larvae have a relatively large head (25% SL) and a moderately deep body (depth 15% SL), which increases to 19% by 4 mm SL. The notochord begins to flex by 5 mm, and flexion is complete by about 5.5 mm SL or shortly thereafter. Rays begin to develop in the dorsal and anal fins of the early postflexion larvae, and by 7.5 mm SL, all fins have a full complement of primary elements. The postflexion larvae lack pigment ventrally on the body, except on the ventral surface of the head and on the dorsal and ventral margins of the pectoral fin base.

3.5.2. Abundance and Spawning Season of *B. pterotum*

The larvae of *B. pterotum* occurred in 43% of all samples taken and accounted for 14.8% of all fish larvae collected at an overall average concentration of 14/1000 m³. The larvae were collected during all months, except September and October, and were the most abundant between November and April, with a peak in January (Figure 5). In fact, January accounted for >50% of all *B. pterotum* larvae collected (255 of 485 larvae). Although not significantly different among bays, abundance was somewhat higher in Naama Bay (180/1000 m³), then Port Bay (161/1000 m³) and Sharm El-Maya Bay (137/1000 m³).

### 3.5.3. Length–Frequency Distribution of *B. pterotum* Larvae

The larvae ranged in size from 2.3 to 7.5 mm SL and were divided into six size classes at one mm length intervals from I: 2–2.9 mm to VI: 6–6.9 mm. Larvae < 5 mm (i.e., size classes I, II, and III) accounted for nearly 75% of all the *B. pterotum* collected, and those between 5 and 6.9 mm for about 25%. The smallest larva was collected during December in Port Bay and the largest larva during January. Based on monthly length–frequency distributions, small larvae were collected every month, except September and October, which suggests that spawning may occur year-round, with peak spawning in January and February, and at much lower levels from late spring to early fall (Figure 10).

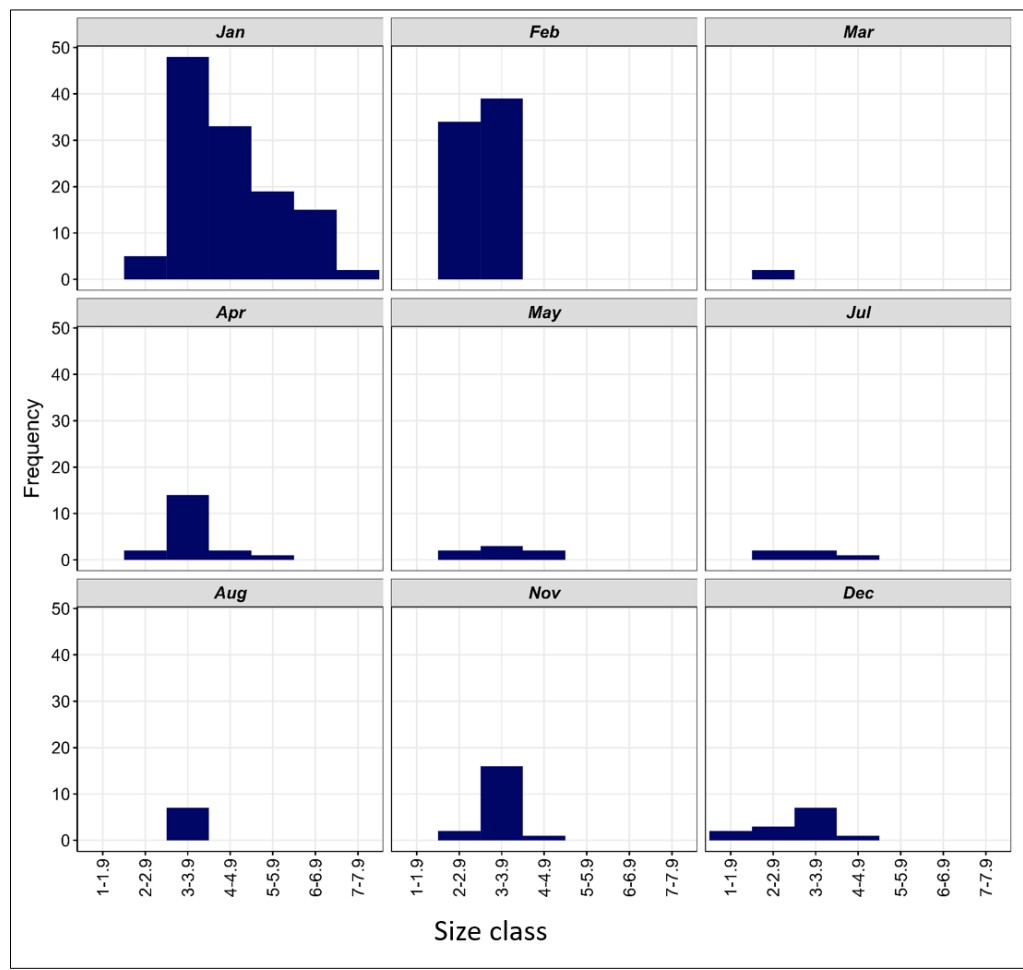

**Figure 10.** Length–frequency distribution of *B. pterotum* from the Sharm El-Sheikh area at different months during the period from January to December 2015.

In August, seven larvae of the same size class III (3–3.9 mm) were collected. In November, the length varied between 2 mm and 4.9 mm, with nearly 93% of all larvae being in size classes I and II (2–3.9 mm). In December, 13 larvae were collected, all <5 mm SL (preflexion). About 54% of these larvae were in size class II (3–3.9 mm). In January, 122 larvae representing all size classes (I–VI) were collected, with about 70% < 5 mm SL. In February, one peak was observed representing size groups I and II with 47% and 53%, respectively. The March sample contained only two larvae, both of similar length. A total of 19 larvae were collected in April and in size were in the range of 2–5.9 mm SL, with the most abundance in size class II (3–3.9 mm). The seven larvae collected in May were in the range of 2–4.9 mm SL and belonged to size classes I, II, and III, as did the five larvae taken in July.

## 4. Discussion

### 4.1. Diversity and Distribution of Mesopelagic Fish Larvae

The Red Sea, an area characterized by high endemism [21], contains 16 species of mesopelagic fishes belonging to nine families and six orders (Table 1). Despite the unusual environmental conditions of its mesopelagic waters (warm, stable water temperatures (~ 22 °C); high salinity (40.5 ppt); and low zooplankton concentrations [22]), the Red Sea reportedly has abundant populations of mesopelagic fishes during the day [21], dominated by *V. mabahiss* at ~500 m [23]. These aggregations of mesopelagic fishes later disperse as they migrate vertically into warmer near-surface waters in the early evening to feed and return to cooler waters before sunrise.

Although the patterns of abundance vary by location, the larvae of *V. mabahiss* and *B. pterotum* dominate the collections of mesopelagic fishes in the Red Sea [16,18,23] and are important components of the ichthyoplankton community [23]. Small *V. mabahiss* larvae were abundant from November to April, with peaks in November and January, and were absent from the collections taken from June to August. By comparison, Isari et al. [23] stated that there was a gradual increase in the abundance of *V. mabahiss* at an offshore station in Suadi, central Red Sea, from January to April, a decline in abundance thereafter through August, and a low secondary peak in September. At the inshore station, monthly estimated abundances were consistently low throughout the year, with a minor peak during June [23]. However, *Vinciguerria mabahiss* was weakly represented and *B. pterotum* was completely absent in a coastal reef lagoon at the Hurghada coast of the Red Sea [24] compared to the current results and previous ichthyoplankton surveys in the area. Differences in sampling methodologies and locations may help to explain this discrepancy in seasonal abundance patterns, because the Gulf of Aqaba waters were sampled off Egypt by horizontal tows within one meter of the sea surface in this study, whereas Isari et al. [23] sampled waters down to 50 m off central Saudi Arabia with oblique tows. Otolith-derived back-calculated hatch dates between June and October for *V. mabahiss* larvae collected in the central Red Sea [25], however, when combined with our findings and those by Isari et al. [23], are consistent with the suggestion that *V. mabahiss* spawn continuously throughout the year [23] at least in some parts of the Red Sea.

Although the larvae of *V. mabahiss* were the most abundant mesopelagic species collected in this study, and the second most abundant species in total abundance among all taxa collected near coral reefs in the Indo-Pacific [26], this was a somewhat surprising result. The island mass effect may help to explain these findings, however, as physical processes near island and other underwater structures may have enhanced the input of nutrients in the euphotic layer, which favors biological productivity, as nutrient-rich deeper waters are brought up towards the surface through upwelling and mixing [27].

The larvae of *Astronesthes martensii* were the least abundant mesopelagic species collected in this study (n = 29 total larvae) and are rare to absent from other ichthyoplankton collections [16,23], as are juveniles and adults in the deep-water trawl collections taken in the southeast Arabian Sea [28]. This finding should not be surprising, as *Astronesthes martensii* typically inhabit waters at depths > 1000 m and ichthyoplankton surveys rarely sample waters at that depth.

The larvae of *B. pterotum* were collected during all months sampled in this study, except September and October, with larvae being the most abundant between November and April, with a peak abundance during January; the larvae were collected at relatively low abundance levels from May to October. By comparison, Isari et al. [23] collected larvae in the smallest size classes during all months, except during July and August. Although the larvae were consistently less abundant in the collections taken by Isari et al. [23] than those in this study, Isari et al. [23] showed a small spike in abundance about every three months throughout the year at their offshore stations. Back-calculated hatch dates for *B. pterotum* in the Sea of Oman suggest a primary spawning season from May to September [29], which is generally consistent with that of *B. pterotum* in the East China Sea that spawn

primarily during August and September [30,31]. Although based on the gonadosomatic index (GSI), a small percentage of mature females occur from May to January based on the gonadosomatic index (GSI) [32]. Dalpadado and Gjesaeter [33] studied the reproduction of *B. pterotum* in the Red Sea and found no direct evidence of a second spawning in the Red Sea, although a second mode of ova was observed in a few females, which is similar to the ones observed by Dalpadado [34] from other areas.

*Benthosema pterotum* eggs are spawned in deep waters (from 200 m to 300 m in the Gulf of Oman); yolk sac larvae are found between 100 and 300 m, and older larvae are generally found between 5 m and 200 m [35]. If a time to hatching of about 10–12 h at a water temperature of 21 °C is common for the Red Sea, larvae hatch at about 1.5 mm SL [30], and if the daily growth rates approach those for *B. pterotum* larvae in the East China Sea (mean absolute growth rate of 0.26 mm; Sassa et al. [31]), the larvae in the two smallest size classes (2–3.9 mm SL) in this study suggest collection within 7 to 10 days of spawning. A mean absolute growth rate of 0.26 mm [31] appears reasonable given an overall mean absolute growth rate of 0.21 mm per day based on the growth rates for eight other species of myctophids (see [31]). Therefore, the estimated hatch date for the largest *B. pterotum* larva collected in this study (7.5 mm SL) is 23–25 days prior to the collection date, which is consistent with an estimated larval duration period of 42 days for the smallest juvenile (12 mm SL) collected by Sassa et al. [31].

### 4.2. The Diagnostic Features of Vinciguerria Species, B. pterotum, and A. martensii

*V. mabahiss*, endemic to the Red Sea and Gulf of Aqaba [36], are one of five species of *Vinciguerria* that have been described at present. *V. mabahiss* is the most similar to its congeners, *V. nimbaria* and *V. lucetia*, from the Indian Ocean and elsewhere, as all three species possess a pair of symphyseal photophores (just behind the mandibular symphysis), which *V. attenuate* and *V. poweriae* lack [36]. *V. mabahiss* differs from *V. nimbaria* and *V. lucetia* by having a total of 58 to 63 photophores (vs. 64–73) and 37 to 38 vertebrae (vs. 39–44, typically 40–41).

All species of *Vinciguerria* larvae have an enlarged pigment medially or near the ventral margin of the caudal fin. *Vinciguerria mabahiss* larvae, as those of *V. nimbaria and V. lucetia*, have pigment near the ventral margin of the tail and pigment above the anal fin base, whereas *V. poweriae* have the caudal pigment medially and lack pigment above the anal fin. *V. attenuata* also has pigment above the gas bladder [36]. *V. mabahiss* larvae lack pigment typically found at three locations in *V. nimbaria and V. lucetia*: postpectoral, anal, and precaudal, whereas the larvae of all three species have pigment in front of the pectoral fin base, along the base of the anal fin, on the tail, and pigment at the base of upper and lower caudal rays [36,37].

Only two species within Myctophidae (*Benthosema pterotum* and *Diaphus coeruleus*) are reported in the Red Sea [38], and all collected in this study were identified as *B. pterotum* based on the pigment patterns and myomere counts. Myctophid larvae are characterized by the presence of an adipose fin and can be distinguished by differences in the number and placement of photophores and the size at which photophores develop. The larvae collected in this study (from 2.3 to 7.5 mm SL) are generally consistent with the description of the smallest *B. pterotum* larva (5.75 mm SL) by Lee et al. [39], who observed pigment on the gut and on the dorsal margin of the hindgut near the vent, as noted in this paper. Lee et al. [39] also suggested that metamorphosis and photophore development is complete and the juvenile life stage attained by 12–13 mm SL in *B. pterotum*. *Benthosema pterotum* larvae can be confused with the larvae of other myctophids due to the elliptical eye in some species, and with some scarids, but scarids have <27 myomeres and a row of melanophores along or beneath the base of the anal fin that typically extend onto the caudal peduncle, which *B. pterotum* lack. The larvae of *A. martensii* are highly diverse and have a great number of morphological specializations; however, no complete developmental series, with transformation specimens, has been described and identifications in the literature are tentative [40].

The larvae of mesopelagic fishes collected during this study from the Red Sea have some common characteristics, including the absence of spines both in fins and the head, the early formed teeth in one or both jaws, the long snout, and the elliptical eyes. The elongated body and elongated gut in *V. mabahiss* and *A. martensii* distinguish them from the larvae of many other families, such as Clupeidae. The trailing gut of *A. martensii* is a very distinctive feature of the species.

## 5. Conclusions

This study contains valuable information about the early stages of mesopelagic fish, which have rarely been described worldwide in general and in the Red Sea in particular. The larvae of *V. mabahiss* and *A. martensii* are described for the first time in the Red Sea in this paper. Moreover, the spawning seasons and grounds of these deep-water fish are described for the first time. The presence of their larvae in large numbers, which outnumber the larvae of coastal fish, indicates that sea currents play a major role in transporting these larvae. However, more studies on the life history of mesopelagic fish are required.

**Author Contributions:** Methodology, M.A.A.E.-R.; Investigation, M.A.A.E.-R. and J.G.D.; Data curation, M.A.A.E.-R.; Writing—original draft, M.A.A.E.-R. and J.G.D.; Writing—review and editing, M.A.A.E.-R. All authors have read and agreed to the published version of the manuscript.

**Funding:** This research work was funded by the Institutional Fund Projects under grant no. (IFPIP 222-150-1443). The authors gratefully acknowledge the technical and financial support provided by the Ministry of Education and King Abdulaziz University, DSR, Jeddah, Saudi Arabia.

**Institutional Review Board Statement:** Samples were collected by plankton net, and as they were only plankton samples, there were no rules to follow.

**Data Availability Statement:** Data are available on request.

**Acknowledgments:** We are grateful to Mohamed Shehata for help in illustrating the early life stages.

**Conflicts of Interest:** The authors declare no conflict of interest.

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
