# Peer review of "Development and Seasonal Variations of the Larvae of Three Mesopelagic Fishes near Coral Reefs in the Red Sea"

_fishes, doi:10.3390/fishes8100500_

Round 1

Reviewer 1 Report

General evaluation:

This work deals with the identification of some mesopelagic fish larvae in the Red Sea, based on the morphology and meristic characters. Title needs to be more assertive. Rational and objectives also need to be improved. Several sentences in the introduction lacks bibliographic support. The sampling windows is unclear. You give different information on abstract and table (January to December 2015 – as written on abstract or August to July - in the remaining MS, including in Table 1 - without mentioning the sampling year).  Please clarify it. Also shortly discuss the fact that sampling have already 8 years old and its consequences for the major findings. Statistics are not appropriate for this kind of data set. Some correlations, namely the ones between SL and other body measurements, are evident (Lapalisse true). Please remove it. It does not make sense in science of the 21st century. Why to do that? All figures and tables captions need to be self-explanatory. Too much figures. Please you can fused some figures (using letters) without losing information (e.g. Fig. 5, 8 and 11; Fig. 7, 9 and 12; Fig. 3, 7 and 10).  Please see the journal guidelines regarding the citation within the text. English language needs a substantial revision. See also my minor comments

Specific evaluation:

Title

It is not representative of the work done. I would suggest something such as “Identification and temporal length frequency distribution of larvae…”

Abstract:

L13-14: “The identification of larvae was based on the author’s experience and other literature”. You need to improve it. It does not make any sense!? Through morphology based on the existent literature?

L15: Change to “preflexion, flexion, and postflexion stages…”

L16: Remove “according to their size”

L25-26: Change to “…may indicate the spawning of the three species took place in the colder times of the year”

Keywords

L27: please do not repeat words already existent on the title. Replace by some related ones. Use a logical sequence order.

Key Contribution

L28: “two mesopelagic fish”!? Two or three?

Introduction

L35: “euphoric”!? or Euphotic?

L39-41: “with a total biomass of billions of tons of which about 300 million tons occur 40 in the Indian Ocean”.

L49-50: “and are important in transferring energy between ocean ecosystems”. Add a supporting reference at the end of this sentence.

L51-57: Add, at least, a supporting reference at the end of this sentence.

L51-52: “and average from a few hundred to few thousand eggs over the spawning season” Please improve English.

L62: Change to “unexplored hialeutic commercial resources

L62-65: Add, at least, a supporting reference at the end of this sentence.

L68-70: “due to the unusually isothermal structure of the waters below the sill level that makes the Red Sea unsuitable for all but a very few adaptable species.” Add, at least, a supporting reference at the end of this sentence.

L71-72: Unclear link between “spawning season” and “morphology”. Re-write

L72-73: “describe the developmental stages of three mesopelagic species” Not correct. You are only dealing with larvae. Re-write.

Table 1: You need to add the references that you used to construct this table.

Material and Methods

L79-91: you need to support the different anthropic profiles of each sampling bay with adequate references.

Fig. 1: Figure captions should be self-explanatory. Meaning of NAM, SMB and PRT?

L113: any explanation to preserve in ethanol 70% and not in 90% as previously mentioned (L103)?

L120-121: “A correlation between the standard length and other measurements was tested by SPSS 22.” Which measurements? Abiotic variables? You did not mention it yet. Please clarify in the MS.

L123-124: “All data were tested for homogeneity of variance”. And about the normal distribution? And which tests did you used it?

L127-129: I would delete the ethical statement. For me it does not make sense here.

Results

L132: Change to “four species belonging to four families and three orders”

L137: “likely”!? This is science.

Table 1: again caption should be self-explanatory (NAM, PRT and SMB). Moreover, on the abstract you mentioned that sampling took place from January to December 2005. But your data starts in Aug? which year?

L150-154: I am not sure if you could use ANOVA for abundance date. But to use a One-way ANOVA to deal with two factor month and bay is not adequate.  

Table 3: Why three of this data have an asterisks. It is unclear to the reader.

Fig.4: This does not make any sense.  This was already expected, bigger larvae, big body measurements. This is not 21st century science.

Fig. 5, Fig. 8 and Fig.11: Add units of measure in Y axis. Again explain the meaning of the abbreviations in the caption.

Fig. 6, Fig. 9 and Fig 12: Add units of measure in X axis

Fig. 3, Fig. 7, Fig. 10, : Please add the author of the picture.

L196. Delete “)”

L199: “month”. Add “s”.Plural

Discussion:

L320: Change to: “…fishes belonging to 9…”

L334-346: Bad English of the first half of this sentence.

L350: Change to “…and the second one…”

L351-355 and L359-359: It is unclear for the reader!? Please re-write.

L380 and 381: Change to “300 m” and “200 m. Add a space)

L386: “within 7 to 10 of spawning”!? Please clarify it.

L396: Change to “elsewhere, as” (add comma).

L399: “without overlap” please rephrase. Bad English.

L402: “complex” or fin?

To be improved

Author Response

Suggestions for Authors

General evaluation:

This work deals with the identification of some mesopelagic fish larvae in the Red Sea, based on the morphology and meristic characters. Title needs to be more assertive. Rational and objectives also need to be improved. Several sentences in the introduction lacks bibliographic support. The sampling windows is unclear. You give different information on abstract and table (January to December 2015 – as written on abstract or August to July - in the remaining MS, including in Table 1 - without mentioning the sampling year).  Please clarify it. Also shortly discuss the fact that sampling have already 8 years old and its consequences for the major findings. Statistics are not appropriate for this kind of data set. Some correlations, namely the ones between SL and other body measurements, are evident (Lapalisse true). Please remove it. It does not make sense in science of the 21st century. Why to do that? All figures and tables captions need to be self-explanatory. Too much figures. Please you can fused some figures (using letters) without losing information (e.g. Fig. 5, 8 and 11; Fig. 7, 9 and 12; Fig. 3, 7 and 10).  Please see the journal guidelines regarding the citation within the text. English language needs a substantial revision. See also my minor comments.

Dear respected reviewer

Thank you very much for the critical revision of our article. Personally, I would like to thank you very much for your effort to improve the quality of the manuscript. Your comments are valuable and correct. We have gone through the manuscript very carefully and all comments were considered and corrected as possible. All figures and table captions have been corrected and the unnecessary sentences were removed. Below are the answers and responses to your other valuable comments

Specific evaluation:

Title

It is not representative of the work done. I would suggest something such as “Identification and temporal length frequency distribution of larvae…”

Response

The title has been changed as per recommendation of the respected reviewer

Abstract:

L13-14: “The identification of larvae was based on the author’s experience and other literature”. You need to improve it. It does not make any sense!? Through morphology based on the existent literature?

Done

L15: Change to “preflexion, flexion, and postflexion stages…”

Done

L16: Remove “according to their size”

Removed

L25-26: Change to “…may indicate the spawning of the three species took place in the colder times of the year”

Changed

Keywords

L27: please do not repeat words already existent on the title. Replace by some related ones. Use a logical sequence order.

The keywords have been changed and re-arranged

Key Contribution

L28: “two mesopelagic fish”!? Two or three?

Corrected

Introduction

L35: “euphoric”!? or Euphotic?

Changed to Euphotic

L39-41: “with a total biomass of billions of tons of which about 300 million tons occur in the Indian Ocean”.

Mesopelagic species are the most abundant marine vertebrates in the world oceans [4] with a total biomass of billions of tons of which about 300 million tons occur in the Indian Ocean

L49-50: “and are important in transferring energy between ocean ecosystems”. Add a supporting reference at the end of this sentence.

Reference was added

L51-57: Add, at least, a supporting reference at the end of this sentence.

L51-52: “and average from a few hundred to few thousand eggs over the spawning season” Please improve English.

Corrected

L62: Change to “unexplored hialeutic commercial resources

L62-65: Add, at least, a supporting reference at the end of this sentence.

References were added

L68-70: “due to the unusually isothermal structure of the waters below the sill level that makes the Red Sea unsuitable for all but a very few adaptable species.” Add, at least, a supporting reference at the end of this sentence.

Robitzch and Berumen, 2020.

Robitzch, Vanessa & Berumen, Michael. (2020). Recruitment of coral reef fishes along a cross-shelf gradient in the Red Sea peaks outside the hottest season. Coral Reefs. 39. 10.1007/s00338-020-01985-9.

L71-72: Unclear link between “spawning season” and “morphology”. Re-write

Done

L72-73: “describe the developmental stages of three mesopelagic species” Not correct. You are only dealing with larvae. Re-write.

Changed to larval stages

Table 1: You need to add the references that you used to construct this table.

Golani and Fricke (2018).

Golani, Daniel & Fricke, Ronald. (2018). Checklist of Red Sea fishes with delineation of the Gulf of Suez, Gulf of Aqaba, endemism and Lessepsian migrants. Zootaxa. 4509. 1-215. 10.11646/zootaxa.4509.1.1.

Material and Methods

L79-91: you need to support the different anthropic profiles of each sampling bay with adequate references.

Reference was added

Fig. 1: Figure captions should be self-explanatory. Meaning of NAM, SMB and PRT?

Done and all acronyms of the bays are added in a separate table

L113: any explanation to preserve in ethanol 70% and not in 90% as previously mentioned (L103)?

The correct concentration is 90% and it was corrected

L120-121: “A correlation between the standard length and other measurements was tested by SPSS 22.” Which measurements? Abiotic variables? You did not mention it yet. Please clarify in the MS.

I mean body length and other morphometrics such as head length…..and it was represented graphically

L123-124: “All data were tested for homogeneity of variance”. And about the normal distribution? And which tests did you used it?

The test was carried out in SPSS using one-way ANOVA

L127-129: I would delete the ethical statement. For me it does not make sense here.

It is required by the journal

Results

L132: Change to “four species belonging to four families and three orders”

L137: “likely”!? This is science.

Changed to possibly

Table 1: again caption should be self-explanatory (NAM, PRT and SMB). Moreover, on the abstract you mentioned that sampling took place from January to December 2005. But your data starts in Aug? which year?

Corrected to Jan-Dec 2015 and all acronyms were added in a separate table

L150-154: I am not sure if you could use ANOVA for abundance date. But to use a One-way ANOVA to deal with two factor month and bay is not adequate. 

One-way ANOVA was used to test differences in abundance among months and then among bays separately> Also, two-way ANOVA was conducted but gave no indications, so it was removed.

Table 3: Why three of this data have an asterisks. It is unclear to the reader.

Deleted

Fig.4: This does not make any sense.  This was already expected, bigger larvae, big body measurements. This is not 21st century science.

You are completely right so the figure was deleted

Fig. 5, Fig. 8 and Fig.11: Add units of measure in Y axis. Again explain the meaning of the abbreviations in the caption.

All were corrected as required

Fig. 6, Fig. 9 and Fig 12: Add units of measure in X axis

Added

Fig. 3, Fig. 7, Fig. 10, : Please add the author of the picture.

These are our own drawings, the drawing was done under the microscope using camera lucida and an artist was hired to polish drawings. A sentence was added to the material and methods section

L196. Delete “)”

L199: “month”. Add “s”.Plural

Discussion:

L320: Change to: “…fishes belonging to 9…”

Done

L334-346: Bad English of the first half of this sentence.

Re-phrased

L350: Change to “…and the second one…”

Corrected

L351-355 and L359-359: It is unclear for the reader!? Please re-write.

Done and the sentence was re-written

L380 and 381: Change to “300 m” and “200 m. Add a space)

Done

L386: “within 7 to 10 of spawning”!? Please clarify it.

The word “days” was missing

L396: Change to “elsewhere, as” (add comma).

Done

L399: “without overlap” please rephrase. Bad English.

Corrected

L402: “complex” or fin?

Corrected

Reviewer 2 Report

In this study, development and length frequency distribution of larvae of some mesopelagic fishes near coral reefs in the Red Sea were assessed. While the study is well designed and conducted, data analysis need a significant revision before publication. Please find my comments below.

Best regards.

Please excuse me If I misunderstood! In line 116, the Authors stated that Abundance of fish larvae was expressed as number of larvae in 1000 m3 based on the following equation: A=N/V. I think this equation would give abundance per unit volume (not per 1000m3) which equals "density" (not abundance). Also according to the table 2, the data are zero-inflated so that the ANOVA test may not be suitable for analysing this type of data. I would suggest Generalized Linear models.

Line 137: The larvae of mesopelagic fishes constituted about 32% of all larvae. Larvae of all fish or Larvae of fish and invertebrates?

Table 2: Are these values means, medians, sums or ...? What do the authors mean by "general abundance"? I strongly suggest that mean values are given in the table

Fig 2 includes replicate charts at the right panel. Please delete one of them.

Line 150: I suggest that the interactive effects of month and site would be tested by ANOVA tests for each species.

Fig 4: in the bottom panel, the X axis title is "body length" which should be "body depth" I presume. Also the title of the figure may be revised to more relevant title (e.g. the correlation between morphometrics of ...)

Line 222: it is not clear when did the V.  mabahiss spawning occur in the area? throughout the year? please revise the text.

Line 226. I think a heading is missing here: (3.3. Astronesthes martensii )

Does the abundance of the three species correlate with each other (either negatively or positively)?

The abundance of post-flexion/ pre-flexion larvae is highly correlated with water temperature. I would suggest to add monthly records of SST in the study area to analyse the possible correlation.

with each other?

N/A

Author Response

Comments and Suggestions for Authors

In this study, development and length frequency distribution of larvae of some mesopelagic fishes near coral reefs in the Red Sea were assessed. While the study is well designed and conducted, data analysis need a significant revision before publication. Please find my comments below.

Best regards.

Please excuse me If I misunderstood! In line 116, the Authors stated that Abundance of fish larvae was expressed as number of larvae in 1000 m3 based on the following equation: A=N/V. I think this equation would give abundance per unit volume (not per 1000m3) which equals "density" (not abundance). Also according to the table 2, the data are zero-inflated so that the ANOVA test may not be suitable for analysing this type of data. I would suggest Generalized Linear models.

Thank you very much for pointing out. Actually, the net was provided with a flowmeter to calculate the volume of water filtered.

V= pr2

where ‘V’ is the volume of water filtered, pr2 is the mouth area of the net in m2, ‘d’ is the distance in meters traveled by the ship, and ‘ƒ’ is the filtration coefficient. The abundance was calculated per m3 but because the number of fish larvae are small, we have to calculate per 1000m3, and we have to transform data into square root to avoid the high number of zeros.

Line 137: The larvae of mesopelagic fishes constituted about 32% of all larvae. Larvae of all fish or Larvae of fish and invertebrates?

Of all larvae, invertebrate larvae were not included

Table 2: Are these values means, medians, sums or ...? What do the authors mean by "general abundance"? I strongly suggest that mean values are given in the table

Mean values

Fig 2 includes replicate charts at the right panel. Please delete one of them.

You are right, the figure was deleted

Line 150: I suggest that the interactive effects of month and site would be tested by ANOVA tests for each species.

Thank you for the suggestion we have already conducted two-way ANOVA and added to the material and method and results sections.

Fig 4: in the bottom panel, the X axis title is "body length" which should be "body depth" I presume. Also the title of the figure may be revised to more relevant title (e.g. the correlation between morphometrics of ...)

Thank you for the note. This figure was removed as per request of one of the reviewers

Line 222: it is not clear when did the V.  mabahiss spawning occur in the area? throughout the year? please revise the text.

Done

Line 226. I think a heading is missing here: (3.3. Astronesthes martensii )

Thank you for the note. Indeed an indenting mistake which was corrected

Does the abundance of the three species correlate with each other (either negatively or positively)?

Thank you very much for this point. Indeed, there was a significant strong relationship between V. mabahiss and B. pterotum and weak and moderate negative relationships between A. martensii from one side and B. pterotum and V. mabahiss from the other side respectively. This statement was added to the result section.

The abundance of post-flexion/ pre-flexion larvae is highly correlated with water temperature. I would suggest to add monthly records of SST in the study area to analyse the possible orrelation.

 with each other?

The relation was tested, and the statement was added to the text

Reviewer 3 Report

The article may be published with minor corrections.

The paper contains valuable new data on the early stages of mesopelagic fish that have a wide oceanic distribution and are also occur in the Red Sea.

The Introduction is well done and provides a compact but informative overview of mesopelagic fishes in general, with a list of 16 Red Sea species. The Methods are described in sufficient detail. The results include, in addition to descriptions of the larvae of the 3 species, some data on their spatial and temporal variability. Larvae of two species are described for the first time in this area. Conclusions are justified.

Minor corrections are suggested in the comments to the text.

Author Response

Comments and Suggestions for Authors

The article may be published with minor corrections.

The paper contains valuable new data on the early stages of mesopelagic fish that have a wide oceanic distribution and are also occur in the Red Sea.

The Introduction is well done and provides a compact but informative overview of mesopelagic fishes in general, with a list of 16 Red Sea species. The Methods are described in sufficient detail. The results include, in addition to descriptions of the larvae of the 3 species, some data on their spatial and temporal variability. Larvae of two species are described for the first time in this area. Conclusions are justified.

Minor corrections are suggested in the comments to the text.

Response

All comments embedded in the attached pdf are very valuable and were considered

The title was changes as per your request

The map has been adjusted by pointing out the study area

Round 2

Reviewer 1 Report

You greatly improve your MS, but I would suggest a few changes:

L14: Change to: "The identification of larvae was based on the morphological and meristic characteristics according with the available literature".

L15: " based on the flexion of the notochord" You could delete this. Is already implicit.

L28: "...and in preflexion stage..."

L30: "here in of which" bad English

L53: "range"

L73: "areare" remove duplication

Table 2: Again. Acronyms meaning NAM, PRT and SMB - needs to be in the caption also. 100 m, 35 m and 10 m (add a space tbetween the number and th iuunits of measure - check through the MS) 

L96: delete "environmantal conditions,"

L119: Again. Two factors, ideally two-way ANOVA. You decided to use two single ANOVA, but like that you missed the interaction.

L122: . "All data were tested for homogeneity of variance and normal distribution" (add "normal distruibution").

L134: bad English - you need to seek language assistance

L158: change to "contribution percentage" 

L164: "Moreover, the interaction between site*month was not statistically 164 significant." You can you say that if you did not perfotmed a two-way ANOVA?

L193: add (personal drawings). same for fig. 6 and 8

Fig 4: again. Add the meaning of the acronyms in the caption!

L238: section 3.3.1 should be below.

You ignore my comments about the fact that your samples are alerady 8 years old. tghis need sto be shothly discussed

Needs professional assistance

Author Response

Comments and Suggestions for Authors

Dear respected reviewer

Thank you very much again for your valuable correction that continue improving our article greatly.

Below are the responses to your comments

You greatly improve your MS, but I would suggest a few changes:

L14: Change to: "The identification of larvae was based on the morphological and meristic characteristics according with the available literature".

Corrected

L15: " based on the flexion of the notochord" You could delete this. Is already implicit.

Right, deleted

L28: "...and in preflexion stage..."

Corrected

L30: "here in of which" bad English

Corrected

L53: "range"

Corrected

L73: "areare" remove duplication

Corrected

Table 2: Again. Acronyms meaning NAM, PRT and SMB - needs to be in the caption also. 100 m, 35 m and 10 m (add a space tbetween the number and th iuunits of measure - check through the MS) 

Done and acronyms were added wherever needed

L96: delete "environmantal conditions,"

Deleted

L119: Again. Two factors, ideally two-way ANOVA. You decided to use two single ANOVA, but like that you missed the interaction.

Two-way ANOVA was conducted to test the interaction between months and sites and it was added to the material and methods section and the result was added to the result section.

L122: . "All data were tested for homogeneity of variance and normal distribution" (add "normal distruibution").

Added

L134: bad English - you need to seek language assistance

The second authors are native -English speaker and he will again go through the manuscript for the language editing

L158: change to "contribution percentage" 

Corrected

L164: "Moreover, the interaction between site*month was not statistically 164 significant." You can you say that if you did not perfotmed a two-way ANOVA?

Two-way ANOVA was conducted to test the interaction between months and sites and it was added to the material and methods section and the interpretation  was added to the result section.

L193: add (personal drawings). same for fig. 6 and 8

A sentence was added to the material and methods section to indicate we , personally, have made the drawing of larvae

Fig 4: again. Add the meaning of the acronyms in the caption!

Done

L238: section 3.3.1 should be below.

Adjusted

You ignore my comments about the fact that your samples are alerady 8 years old. tghis need sto be shothly discussed

The samples were collected in 2015 but we were not able to work on them at that time. Because most of the work is taxonomic, we think there will no be big difference in the results.

Comments on the Quality of English Language

Needs professional assistance

The second authors are native -English speaker and he will again go through the manuscript for the language editing

Submission Date

10 September 2023

Date of this review

01 Oct 2023 13:56:52

Reviewer 2 Report

I suggest the manuscript can now be accepted for publication now.

Author Response

Comments and Suggestions for Authors

Dear respected reviewer

Thank you very much again for your valuable correction that continue improving our article greatly.

Below are the responses to your comments

You greatly improve your MS, but I would suggest a few changes:

L14: Change to: "The identification of larvae was based on the morphological and meristic characteristics according with the available literature".

Corrected

L15: " based on the flexion of the notochord" You could delete this. Is already implicit.

Right, deleted

L28: "...and in preflexion stage..."

Corrected

L30: "here in of which" bad English

Corrected

L53: "range"

Corrected

L73: "areare" remove duplication

Corrected

Table 2: Again. Acronyms meaning NAM, PRT and SMB - needs to be in the caption also. 100 m, 35 m and 10 m (add a space tbetween the number and th iuunits of measure - check through the MS) 

Done and acronyms were added wherever needed

L96: delete "environmantal conditions,"

Deleted

L119: Again. Two factors, ideally two-way ANOVA. You decided to use two single ANOVA, but like that you missed the interaction.

Two-way ANOVA was conducted to test the interaction between months and sites and it was added to the material and methods section and the result was added to the result section.

L122: . "All data were tested for homogeneity of variance and normal distribution" (add "normal distruibution").

Added

L134: bad English - you need to seek language assistance

The second authors are native -English speaker and he will again go through the manuscript for the language editing

L158: change to "contribution percentage" 

Corrected

L164: "Moreover, the interaction between site*month was not statistically 164 significant." You can you say that if you did not perfotmed a two-way ANOVA?

Two-way ANOVA was conducted to test the interaction between months and sites and it was added to the material and methods section and the interpretation  was added to the result section.

L193: add (personal drawings). same for fig. 6 and 8

A sentence was added to the material and methods section to indicate we , personally, have made the drawing of larvae

Fig 4: again. Add the meaning of the acronyms in the caption!

Done

L238: section 3.3.1 should be below.

Adjusted

You ignore my comments about the fact that your samples are alerady 8 years old. tghis need sto be shothly discussed

The samples were collected in 2015 but we were not able to work on them at that time. Because most of the work is taxonomic, we think there will no be big difference in the results.

Comments on the Quality of English Language

Needs professional assistance

The second authors are native -English speaker and he will again go through the manuscript for the language editing